# A substrate-based ontology for human solute carriers

Eva Meixner[1], Ulrich Goldmann[1] (ID), Vitaly Sedlyarov[1], Stefania Scorzoni[1], Manuele Rebsamen[1], Enrico Girardi[1,*] (ID) & Giulio Superti-Furga[1,2,**] (ID)

## Abstract

Solute carriers (SLCs) are the largest family of transmembrane transporters in the human genome with more than 400 members. Despite the fact that SLCs mediate critical biological functions and several are important pharmacological targets, a large proportion of them is poorly characterized and present no assigned substrate. A major limitation to systems-level de-orphanization campaigns is the absence of a structured, language-controlled chemical annotation. Here we describe a thorough manual annotation of SLCs based on literature. The annotation of substrates, transport mechanism, coupled ions, and subcellular localization for 446 human SLCs confirmed that ~30% of these were still functionally orphan and lacked known substrates. Application of a substrate-based ontology to transcriptomic datasets identified SLC-specific responses to external perturbations, while a machine-learning approach based on the annotation allowed us to identify potential substrates for several orphan SLCs. The annotation is available at https://opendata.cemm.at/gsflab/slcontology/. Given the increasing availability of large biological datasets and the growing interest in transporters, we expect that the effort presented here will be critical to provide novel insights into the functions of SLCs.

**Keywords** annotation; de-orphanization; ontology; SLCs; solute carriers
**Subject Categories** Membranes & Trafficking; Methods & Resources; Pharmacology & Drug Discovery
**Mol Syst Biol. (2020) 16: e9652**

## Introduction

Solute carrier (SLC) proteins are a large family of multi-pass membrane transporters, counting 446 members divided in 70 families. SLCs are responsible for the transport of a wide range of substrates, including nutrients, ions, and waste products, across plasma membrane and organellar membranes (Hediger *et al*, 2013; César-Razquin *et al*, 2015). Several family members have been the subject of extensive pharmacological research as both drug targets and for their role in drug disposition and toxicology (Lin *et al*, 2015). Moreover, several drugs, including metabolite-like compounds, "hitchhike" transporters of endogenous substrates, a concept that can be exploited for tissue-specific drug delivery (Kell, 2016).

Despite the clear biological relevance of this family, it is estimated that approximately one-third of SLCs are still orphan, i.e., they lack known substrates or metabolic function (Perland & Fredriksson, 2017). This realization has sparked calls and efforts to systematically study and functionally characterize SLCs (Hediger *et al*, 2013; César-Razquin *et al*, 2015; Superti-Furga *et al*, 2020). With mounting interest in this family, it becomes pivotal to define the current limits and gaps of our understanding of SLCs and organize our knowledge in a formally consistent annotation that can be used to interrogate and integrate increasingly large biological datasets.

Ontologies are a powerful tool for the annotation and organization of knowledge in biology and biomedicine. They provide a controlled, hierarchical vocabulary for a specific domain of knowledge, defining terms and the relationships between them and facilitating findability and interoperability between databases and data integration (Lambrix *et al*, 2007; Hoehndorf *et al*, 2015). Moreover, they enable computational methods to systematically investigate biological functions and processes, as in the case of metabolic functions of proteins. Protein function prediction methods typically rely on the identification of homologous genes/proteins or use supervised machine-learning algorithms trained with protein sequence or structure information (Cruz *et al*, 2017).

Here we describe the creation of a manually curated annotation of human SLCs defining their known substrates, transport mechanisms, and subcellular localizations as well as the use of this information to develop a substrate-based ontology of SLCs, which we applied to identify patterns in SLC expression from biological datasets. We further employed this substrate annotation to train a machine-learning model to predict substrates for orphan SLCs. An interactive version of the annotation is available at https://opendata.cemm.at/gsflab/slcontology/.

1 CeMM Research Center for Molecular Medicine of the Austrian Academy of Sciences, Vienna, Austria
2 Center for Physiology and Pharmacology, Medical University of Vienna, Vienna, Austria
*Corresponding author. Tel: +43 1 40160 70001; E-mail: egirardi@cemm.oeaw.ac.at
**Corresponding author. Tel: +43 1 40160 70001; E-mail: gsuperti@cemm.oeaw.ac.at

# Results

### A manually curated SLC annotation

With the goal of systematically characterizing human SLCs, we manually annotated substrates, coupled ions, transport mechanism, and subcellular localization of 446 human SLCs with data derived from primary literature (Table EV1). Depending on the category, 25–51% of the SLCs have no known annotation (Fig 1A), highlighting the large gaps in our knowledge about this protein family. We found that for 126 SLCs no substrates were experimentally confirmed to be transported in human cells yet (28%, including two accessory proteins), which is in line with the estimates of the number of orphan SLCs described in literature (César-Razquin *et al*, 2015; Perland & Fredriksson, 2017). For the remaining 320 SLCs, we annotated a total of 382 different transported molecules, which we divided into substrates and coupled ions (Fig 1B and C), defining coupled ions as ions whose gradients drive transport of the substrates across the membrane (secondary active transport). The mechanism of transport (symporter, antiporter, or uniporter) was also annotated, if known (Fig 1D). In a first attempt to define substrate-based subgroups, we assigned the SLCs to ten different substrate classes, comprising major classes of biomolecules (Fig 1B) as well as a class for orphan transporters. Exactly one class was assigned per SLC, and substrates not belonging in any of those classes were summarized as class "Other". This classification confirmed that SLCs are involved with the transport of a broad range of chemically diverse molecules, from amino acids to vitamins and lipids. In particular, ions and amino acids were the most populated classes with 81 and 54 members, respectively. Sodium, chloride, and protons accounted for approximately half of the coupled ions, while 39 transporters were annotated as uniporters. Individual cell lines express 150–250 SLCs, in patterns similar to the tissue of origin (César-Razquin *et al*, 2018; O'Hagan *et al*, 2018), with members on virtually all intracellular membranes, as well as the plasma membrane. We found that nearly two-thirds of annotated SLCs have been reported to be localized, at least partially, on the plasma membrane. The most frequently assigned annotation of intracellular localization was mitochondria, with 61 mitochondrial SLC transporters (Fig 1E).

Overall, this manual annotation provided subcellular localizations, transport mechanism, and a large number of different transported entities for human SLCs, while highlighting the fact that for many transporters complete information is still missing.

### A substrate-based SLC ontology

The manual annotation assigned at least one known cargo, i.e., substrate or coupled ion (in the following text referred to as "substrate") to 320 SLCs. However, out of 382 substrate terms, only 23 terms are shared by at least 10 SLCs. This poor overlap in substrate annotation of different SLCs made the annotation unsuitable for enrichment analysis in high-throughput datasets and underlined the urgency for knowledge standardization and organization. The Chemical Entities of Biological Interest (ChEBI) ontology contains more than 46,000 manually curated entries, each of them with assigned annotations, synonyms, chemical structure, and database and literature links. By mapping the substrates from manual

annotation to ChEBI ontology terms (Table EV2), a controlled, hierarchical vocabulary for substrate terms was introduced, which included a large number of more general substrate terms with increased overlap between substrate annotations of different transporters. 2,266 ontology terms from the ChEBI ontology were connected to annotated SLC substrates (Fig 2). In order to reduce the number of terms, we filtered out redundant or irrelevant ontology terms by application of two major reduction steps to the ontology graph (Fig 2A): In the ChEBI ontology, chemical tautomers and conjugate bases/acids (protonated and deprotonated forms of the same molecule) are connected by circular relationships. By merging terms connected by those relationships, we could remove 494 terms and 1,516 relationships, making the ontology graph a directed, acyclic graph. Moreover, we removed terms with only one sub-term from the graph, as they represent additional levels of specificity that do not provide further distinction between instances of this branch. Another 742 ontology terms were thus removed, resulting in a final reduced ontology of 1,030 terms connected by 3,458 relationships (without SLCs) (Figs 2B–D and EV1, Tables EV1 and EV3). The chemical sub-ontology consists of 818 terms, whereas the role sub-ontology was reduced to 212 terms (Fig 2E).

### Application of the substrate ontology for term enrichment analysis in transcriptomic data

Next, we wanted to assess the performance of the SLC substrate ontology in identifying patterns within SLC expression values obtained from real-world datasets. We therefore measured transcriptomics profiles of HEK293T cells upon single amino acid deprivation as well as obtained published microarray-based transcriptional profiles of MCF7 breast cancer cells (Tang *et al*, 2015). Extracellular availability of amino acids influences cellular metabolism, while the specific lack of amino acids causes activation of the amino acid response (AAR) pathway (Palii *et al*, 2009). We therefore expected to see changes in SLC gene expression upon amino acid starvation, as the cell attempts to compensate for the depletion of specific nutrients. Depletion of methionine triggered a particularly large change in the global transcription profile in both cell lines (Fig EV2A–D). Depletion of methionine also led to the highest number of upregulated SLC genes (Fig EV2B and D).

Applying our SLC substrate ontology to the set of upregulated genes, a significant enrichment of various amino acid-related substrate terms was detected (Figs 3A and B, and EV3). Interestingly, depletion of several essential amino acids (including leucine, tryptophan, and threonine) did not lead to broad enrichment within the upregulated genes in MCF7 cells but showed strong enrichments in HEK293T cells (Fig 3A and B). Transporters linked to methionine uptake were significantly enriched within the significantly upregulated SLCs during deprivation of cysteine, glutamine, arginine, isoleucine, and histidine in MCF7 cells. Methionine has been shown to act as growth signal in different organisms, inducing proliferation even under nutrient-limiting conditions (Cavuoto & Fenech, 2012; Sutter *et al*, 2013; Walvekar *et al*, 2018). To benchmark the performance of the SLC substrate ontology against the most commonly used alternative, we performed a term enrichment analysis within the upregulated gene set using Gene Ontology (GO) annotations (The Gene Ontology Consortium, 2019). "Amino acid transmembrane transport activity" was significantly enriched for several

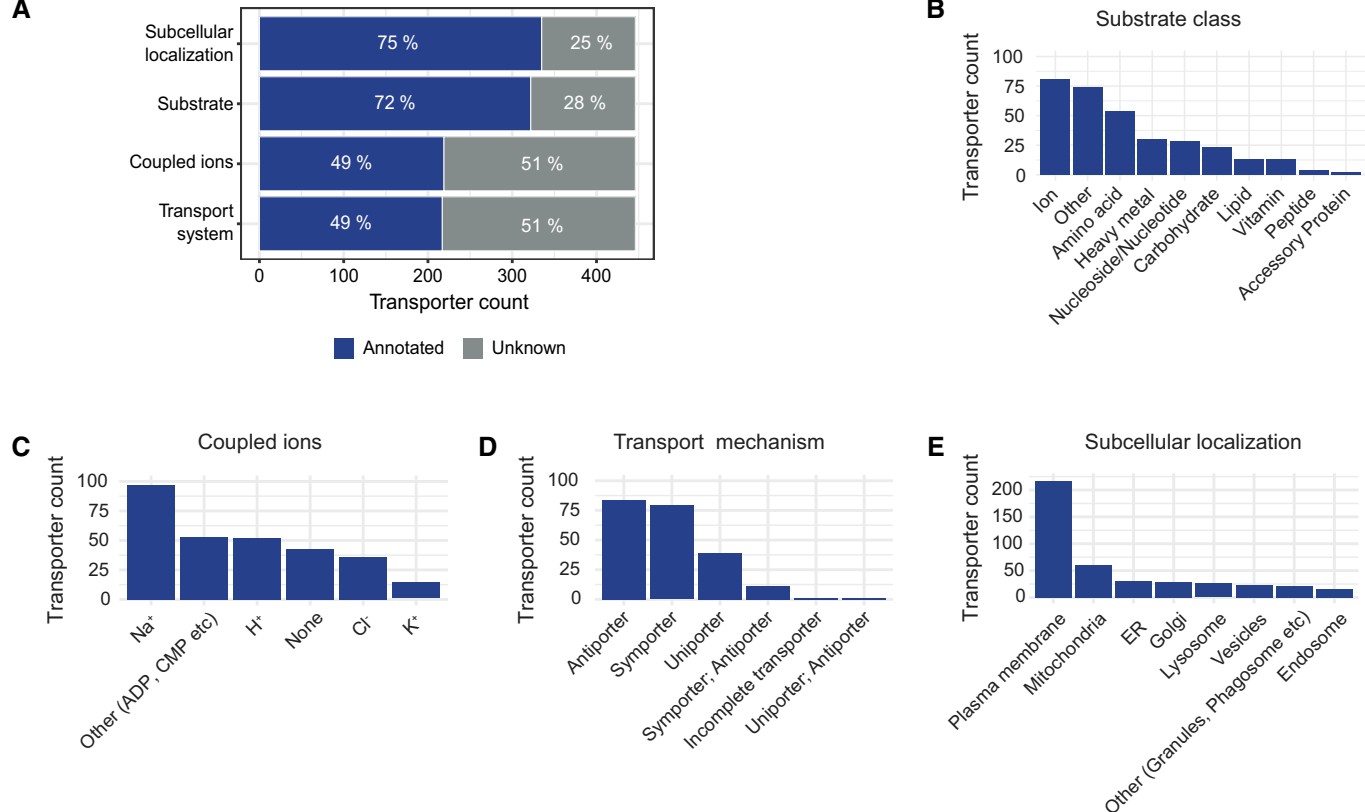

**Figure 1. Manually curated annotation of 446 human SLC transporters.**

A   Frequencies of unknown annotations for 446 SLCs in four annotation categories.

B–E   Distribution of annotated terms for substrate class, coupled ions, transport mechanism, and subcellular localization. Terms annotated to less than ten SLCs were summarized as "Other" in (C) and (E).

different conditions (Fig 3A and B), a lower (for MCF7) or equivalent (for HEK293T cells) number than the ones identified using the SLC ontology. Moreover, the SLC ontology provided a finer distinction via the specific amino acid sub-terms that were enriched in the set of upregulated SLCs.

Overall, the SLC ontology allowed detection of upregulation of amino acid transporters under amino acid starvation conditions across different cell lines and delivered a more precise classification of results than GO.

**Substrate predictions for orphan SLCs**

A major challenge in the solute carrier field remains assigning substrates for the nearly 30% of human SLCs that are still entirely orphan. We trained a machine-learning model to predict probabilities of selected substrate terms for the 124 orphan SLCs in our set using the systematic substrate annotations previously described. We characterized SLCs using protein sequence-derived, numeric features, including sequence and physicochemical properties for four annotated and predicted topological domains: the cytoplasmic domain, the non-cytoplasmic domain, the transmembrane domain, and the signal peptide (see Materials and Methods). Those features were used to train independent random forest binary classifiers for

18 selected substrate terms (Table EV4). Resulting classifiers were found to have high predictive performance, in particular for metal cation and sub-terms (Fig 3C and D). All classifiers were then employed to predict probability of substrate terms for orphan SLCs (Figs 3E and EV4). In many cases, the predictions matched the expected substrate(s) based on family or orthologues in other species. For example, SLC39A11, which in mouse acts as a zinc importer (Yu *et al*, 2013), was also predicted to transport zinc with the highest score within our set. SLC6 family members transport amino acids, neurotransmitters or creatine (SLC6A8) and are all annotated to have sodium, and most of them also chloride, as coupled ions. Accordingly, the orphan transporters SLC6A16 and SLC6A17 were both predicted to transport sodium ions, with the latter also predicted to transport amino acids. Among the large number of associations identified, we further predicted an association with divalent metal cations for the transporters TMEM165, reported to be involved in $Ca^{2+}$ homeostasis (Demaegd *et al*, 2013), NIPAL3—possibly involved in $Mg^{2+}$ transport, as well as the full orphan SLC35F6. Finally, we predicted nucleobase-containing substrates for the mitochondrial transporter SLC25A45 as well as for SLC22A25 and SLC35E2B. Overall, our approach provides a large set of experimentally testable, novel SLC substrate associations for previously orphan SLCs.

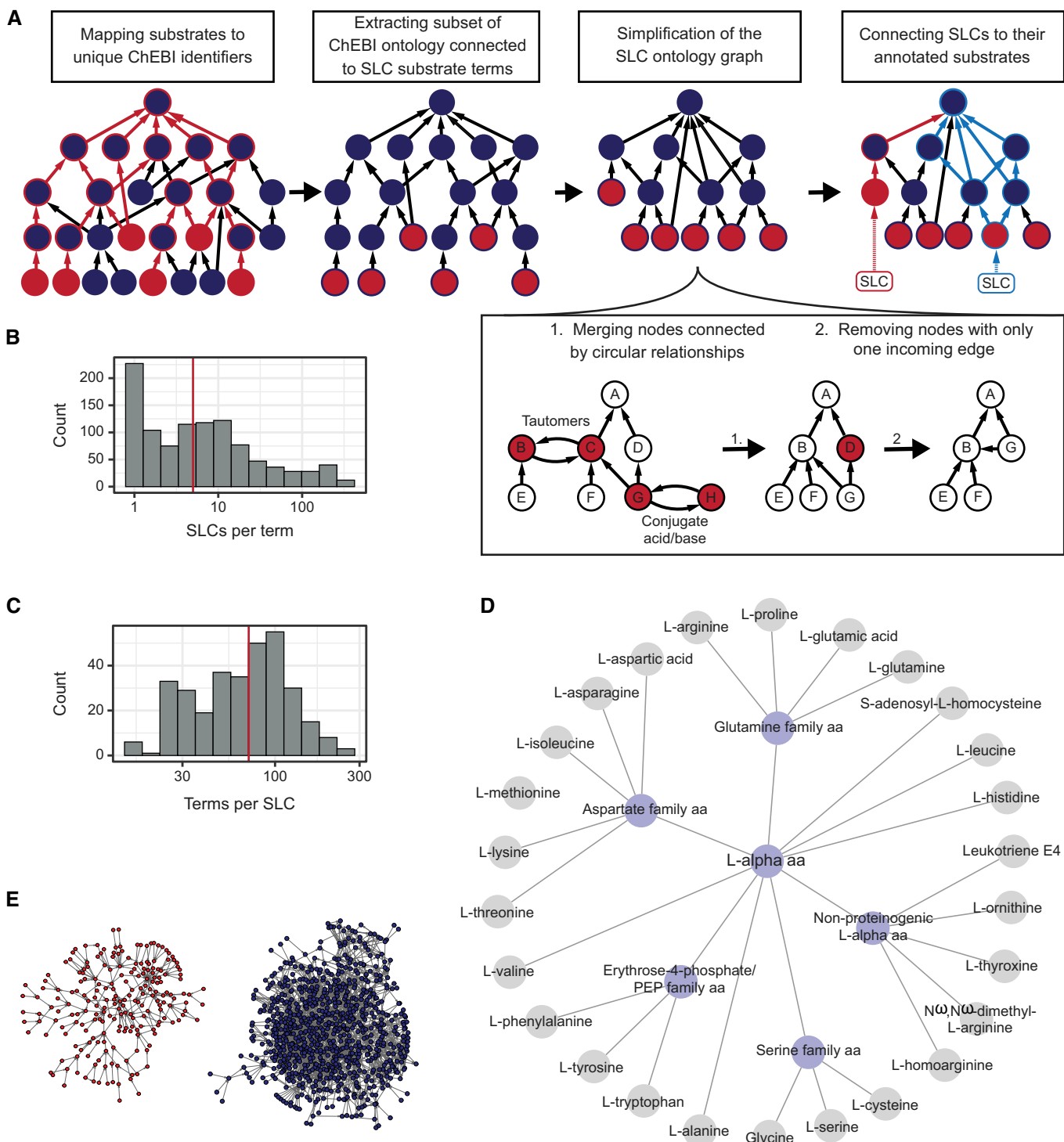

Figure 2. Construction of an SLC substrate-specific ontology from ChEBI ontology.

A  Individual steps in ontology creation workflow.
B  Distribution of the number of SLCs per ontology term. Red line indicates the median value.
C  Distribution of the number of ontology terms associated with one SLC. Red line indicates the median value.
D  Exemplified visualization of term "L-alpha amino acid (aa)" and its sub-terms. This is a sub-graph and SLC substrates (gray) are connected to more terms in the full ontology. Please refer to Fig EV1 for an extended example of the term "amino acid" and its sub-terms.
E  Visualization of the resulting SLC-specific ontology: role sub-ontology (left) and chemical entity sub-ontology (right).

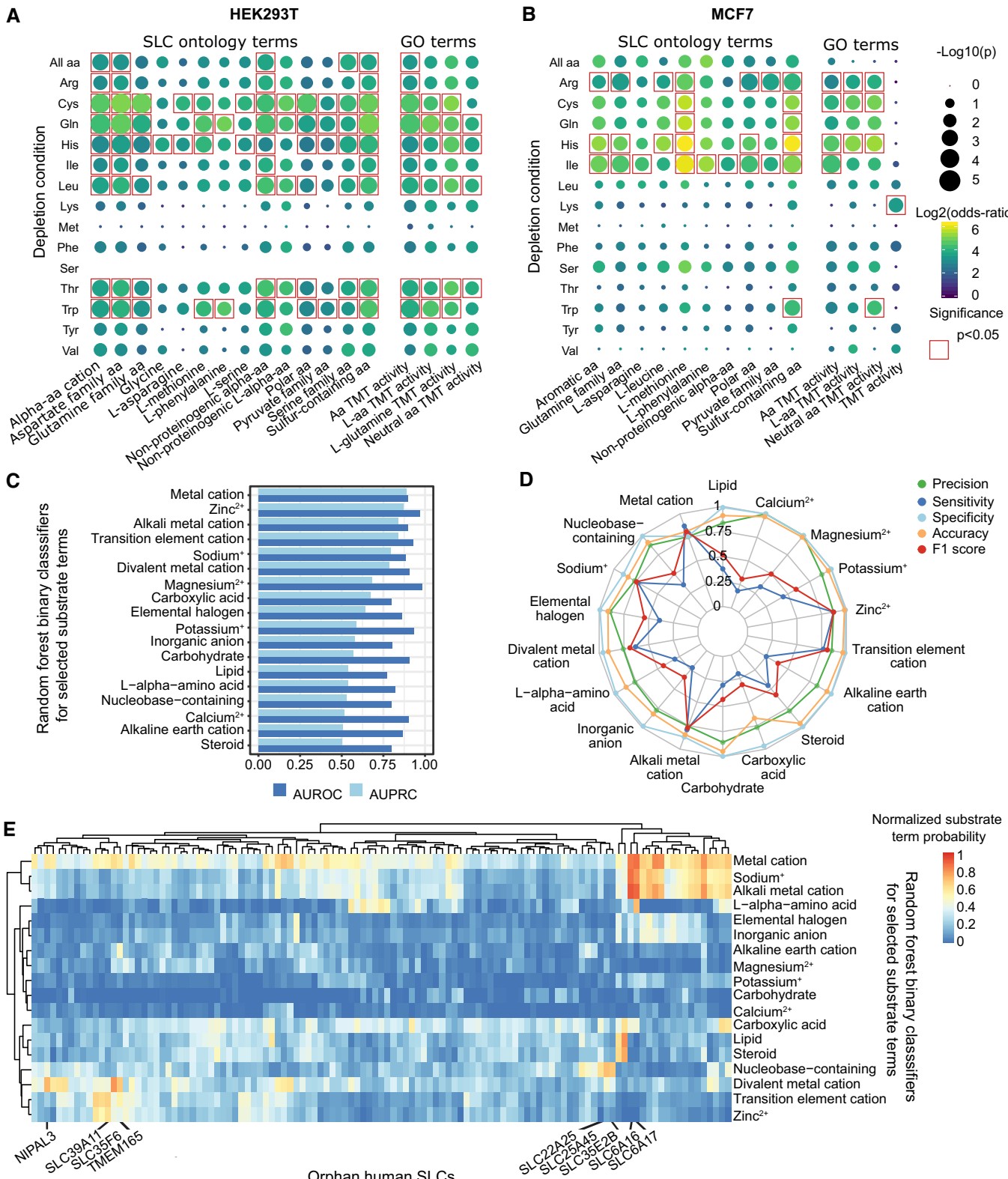

**Figure 3.**

**Figure 3.   Application of SLC substrate ontology and prediction of substrates for orphan SLCs.**

A, B   Ontology term enrichment analysis in set of SLCs upregulated in (A) HEK293T and (B) in MCF7 cells after amino acid (aa) deprivation conditions using SLC ontology terms and GO terms (TMT: transmembrane transporter). Enrichments were calculated using Fisher's exact test. For simplification, only enrichment of the most specific terms is shown in (A,B), for complete version see Fig EV3.

C       Area under the receiver operating characteristic (AUROC) and under the precision recall curve (AUPRC) derived from out-of-bag (OOB) error estimates for random forest classifiers for the 18 selected SLC substrate terms.

D       Statistical performance measures for the binary classifiers from OOB estimates.

E       Predicted substrate term probabilities for orphan SLCs, normalized to a decision threshold of 0.5.

# Discussion

Computational methods for the analysis and interpretation of high-throughput omics data in biology rely heavily on the availability of well-structured metadata complementing the increasingly large experimental datasets available. Existing classifications of SLCs, such as the HUGO Gene Nomenclature Committee (HGNC), the classification proposed in Schlessinger *et al* (2010), and the Transporter Classification Database (TCDB), are based on phylogenetic, structural or functional information but lack systematic substrate annotation. In order to address this, we created a systematic SLC annotation for substrate, transport mechanism, and subcellular localization, as well as an ontology based solely on substrates transported by human SLCs. As SLCs mediate transport of essential nutrients into cells or subcellular compartments, acting as interface between cells and their environment, it follows that it should be possible to infer a cell metabolic state from their activity profile. The SLC substrate ontology described here guides interpretation and comparison of SLC expression levels across different cell lines and upon perturbations, allowing the identification of interesting patterns and transporter-substrate associations worthy of further investigation. Regular updates to the annotations are expected to follow, incorporating newly published data, possibly in conjunction with other SLC-focused resources already available such as the knowledgebases developed by Bioparadigms (www.bioparadigms.org) or ReSOLUTE (www.re-solute.eu).

In conclusion, we are convinced that, given the growing efforts and interest in SLCs and the ever-increasing availability of datasets, SLC-specific annotations and ontologies such as the ones described here will be an essential part of the toolbox required for the systematic understanding of SLC function and organization on a cellular-, tissue- and organismal-level as well as for the de-orphanization of SLCs by orthogonal methods (Kory *et al*, 2018; Yee *et al*, 2019). Moreover, in due time, this annotation will empower the regular enlargement of metabolic charts to encompass the cognate transporters, allowing a cellular and spatial dimension to metabolism to be further captured. Ultimately, one can imagine this to represent a small but important contribution to modeling pathophysiology.

# Materials and Methods

### Manual annotation of SLCs

Substrates, coupled ions, transport mechanism, and subcellular localization for each of the 446 human SLCs were manually annotated from the primary literature. Substrates were defined as molecules showed to be transported by the SLC in a transport assay using reconstituted protein or gene overexpression experiments in human cells. 11 substrate classes were defined: accessory protein, amino acid, carbohydrate, ion (defined as charged small molecules or atoms), lipid, heavy metal ($Cd^{2+}$, $Co^{2+}$, $Cu^{2+}$, $Fe^{2+}$, $Mn^{2+}$, $Ni^{2+}$, $Pb^{2+}$, $V^{3+}$, $VO^{2+}$, $Se^{2+}$, $Zn^{2+}$), nucleoside/nucleotide, orphan, peptide, vitamin and other (the latter including all substrates not fitting in one of the previous categories). For symporters or antiporters, coupled ions were defined as small ions required for the transport of substrates. Subcellular localization was taken from primary literature, when immunofluorescence data obtained in human cells were available. Whenever conflicting results were reported, precedence was given to localization data obtained by co-staining with known organelle markers.

### Construction of SLC substrate ontology

SLC substrate terms (obtained from merging the substrate and coupled ion entries) from the manual annotation were mapped to ChEBI identifiers with Ontology Lookup Service (OLS) from the European Bioinformatics Institute (EMBL-EBI)(Jupp *et al*, 2015). The search was conducted using exact term matching, extracting ChEBI identifiers, labels, and term description for every hit. Terms without match were additionally translated with The Chemical Translation Service (CTS) (Wohlgemuth *et al*, 2010). For cases where both automatic mappings could not find matching entries, ChEBI entries were manually selected (Table EV2).

The ChEBI ontology was downloaded in OBO format ("Chebi_core.obo," September 2019) from the ChEBI portal (ftp://ftp.ebi.ac.uk/pub/databases/chebi/ontology/). The oboe R package was used for parsing of the ontology (Lindholm, 2019). A subset of the ChEBI ontology related to SLC substrates was extracted, by only considering terms reachable within the ontology from the SLC substrate nodes via directed paths. For the extraction, only five types of relationships between terms ("is a," "has role," "is tautomer of," "is conjugate acid of," and "is conjugate base of") were considered and other types of relationships were ignored. Terms belonging to ChEBI's subatomic particle sub-ontology were excluded from the analysis.

The number of ontology terms was reduced by two steps (Fig 2A): At first, terms connected by circular relationships ("is tautomer of" or "is conjugate acid of"/"is conjugate base of") were aggregated into one term. Secondly, non-substrate terms with exactly one incoming edge were removed from the graph. In order to maintain the original connectivity, a new relationship is introduced between parent and child terms of the removed terms.

In a final step, SLC terms are added to the ontology graph. The newly defined type of ontology relationship, "transports," connects SLC terms to chemical entity terms. Ontology terms annotating a

specific SLC are defined as all ontology terms reachable from the SLC term via directed paths.

### Published amino acid starvation data

Gene expression intensities for MCF7 cells after single amino acid starvation were obtained from ArrayExpress (E-GEOD-62673) (Tang *et al*, 2015). The microarray data was processed, and differential expression determined as described in the original publication. SLCs differentially expressed at 5% FDR at the 24 h time point were determined for all conditions.

### Transcriptomics

HEK293T were obtained from ATCC and their identity and lack of contamination with mycoplasma confirmed by STR profiling and PCR, respectively. Cells were seeded in triplicate in full media (DMEM Gibco, 10% FBS Gibco, antibiotics). After 24 h, media were removed and, after a wash with PBS, substituted with DMEM starvation media lacking the indicated amino acid supplemented with 10% (v/v) dialyzed FBS (Gibco cat. 26400-044). Starvation media each lacking a single amino acid were prepared by complementing amino acid-free DMEM media (i.e. devoid of all 15 amino acids normally present, custom made by PAN Biotech) with the other 14 amino acids (from individual amino acid powders, SIGMA). DMEM media reconstituted with all 15 amino acids and 10% dialyzed FBS as well as full media served as controls. After 16 h, media were removed, and cells were harvested in cold PBS. Total RNA was isolated using the Qiagen RNeasy Mini kit including a DNase I digest step. RNA-sequencing (RNA-seq) libraries were prepared using QuantSeq 3′ mRNA-Seq Library Prep Kit FWD for Illumina (Lexogen) according to the manufacturer's protocol. Libraries were subjected to 50-bp single-end high-throughput sequencing on an Illumina HiSeq 4000 platform at the Biomedical Sequencing Facility (https://biomedical-sequencing.at/). Raw sequencing reads were demultiplexed, and after barcode, adaptor, and quality trimming with cutadapt (https://cutadapt.readthedocs.io/en/stable/), quality control was performed using FastQC (http://www.bioinformatics.babraham.ac.uk/projects/fastqc/). The remaining reads were mapped to the GRCh38/h38 human genome assembly using genomic short-read RNA-seq aligner STAR version 2.567. We obtained more than 98% mapped reads in each sample with 70–80% of reads mapping to unique genomic location. Transcripts were quantified using End Sequence Analysis Toolkit (ESAT)(Derr *et al*, 2016). Differential expression analysis was performed using three biological replicates with DESeq2 (1.21.21) on the basis of read counts (Love *et al*, 2014). Exploratory data analysis and visualizations were performed in R-project version 3.4.2 (Foundation for Statistical Computing, https://www.R-project.org/) with Rstudio IDE version 1.0.143, ggplot2 (3.0.0), dplyr (0.7.6), readr (1.1.1), gplots (3.0.1).

### Enrichment analysis

Ontology term enrichment was conducted using one-sided Fisher's exact test. Benjamini–Hochberg procedure was used for multiple testing correction (5% FDR cutoff). Testing was done for upregulated ($\log_2$-fold change > 0.5, 5% FDR) SLCs.

For enrichment of SLC substrate ontology terms, only terms from the "chemical entity" sub-ontology, which are associated with at least five different but < 70% of annotated SLCs, were tested. Results of the enrichment test were simplified by removing redundant terms: For every condition, only the most specific enriched terms of a branch were selected: e.g. if "sulfur-containing amino acid" and "L-methionine" are both enriched, only "L-methionine" is selected. The union of the most specific terms from all conditions was used for plotting (Fig 3A and B); the unfiltered results are shown in Fig EV3A and B. GO terms were assigned to SLCs using the R BiomaRt package (Durinck *et al*, 2005, 2009). Enrichment tests were done separately for the "Biological process" sub-ontology.

### SLC substrate classifiers

Features for every SLCs were derived from protein sequences as previously described (Bausch-Fluck *et al*, 2018). Briefly, protein sequences were obtained from UniProt (Bateman *et al*, 2015) and divided into topological domains according to existing annotations and predictions. Features including amino acid frequencies, glycosylation sites, existence of motifs, and average length were defined separately for four domains (cytoplasmic, non-cytoplasmic, transmembrane region, signal peptide).

Random forest binary classifiers were trained using the randomForest R package version 4.6.14 (Liaw & Wiener, 2003). In total, 18 substrate terms were selected from the SLC substrate ontology which are generic enough to have a sufficient number of annotated SLCs for training as well as specific enough to allow hypothesis generation by subsequent prediction on orphan SLCs. For each substrate term, an independent, binary classifier was trained on the set of 304 SLCs with known substrates and sequence features, using those SLCs with a known substrate matching to the specific ontology term as a positive training set and the remaining SLCs with a known substrate as a negative training set. In total, 18 binary classifiers for different substrate classes derived from the SLC substrate ontology were trained on the set of 304 SLCs with known substrates and sequence features. The parameters mtry (20–200), ntree (300–1,500), and classwt (unweighted or class priors) were optimized using a grid search maximizing F1-score for every classifier separately. Decision thresholds were set to the threshold that gives maximum recall for a precision value of at least 75%. Predicted scores were normalized to have a decision threshold of 0.5 by gamma correction. Predictive performance of the classifiers was estimated using "out-of-bag" estimates provided by the random-Forest library.

## Data availability

The HEK293T transcriptomics dataset is deposited at GEO (GSE153034). Datasets are provided in Tables EV1–EV4 and are also available at the accompanying web site at https://opendata.cemm.at/gsflab/slcontology/.

**Expanded View** for this article is available online.

## Acknowledgements

We are grateful for the contribution of several present and former members of the Superti-Furga laboratory to the manual annotation of subsets of SLCs: Ariel Bensimon, Johannes W. Bigenzahn, Vojtech Dvorak, Ruth Eichner, Patrick Essletzbichler, Giuseppe Fiume, Leonhard Heinz, Alvaro Ingles-Prieto, Vasyl Ivashov, Felix Kartnig, Kai-Chun Li, Sabrina Lindinger, Tea Pemovska, Mattia Pizzagalli, Ismet Srndic, Nadine Tuechler, Tabea Wiedmer and Gernot Wolf. We acknowledge support by the Austrian Academy of Sciences, the European Research Council (ERC AdG 695214 GameofGates) and the Innovative Medicines Initiative 2 Joint Undertaking (grant agreement No 777372, ReSOLUTE).

## Author contributions

Conceptualization: UG, VS, GS-F and EG; Ontology generation and dataset analyses experiments: EM; Data and annotation analysis: EM, UG, VS, and EG; Design and experiments: MR; experiments and transcriptomics experiments: SS; Supervision: EG and GS-F. All authors contributed to the manuscript preparation.

## Conflict of interest

The authors declare that they have no conflict of interest.

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
