## [Review Process File · Molecular Systems Biology]

A substrate-based ontology for human Solute Carriers

Giulio Superti-Furga, Eva Meixner, Ulrich Goldmann, Vitaly Sedlyarov, Stefania Scorzoni, Manuele Rebsamen, and Enrico Girardi

DOI: [10.15252/msb.20209652](https://doi.org/10.15252/msb.20209652)

Corresponding author(s): Giulio Superti-Furga (gSuperti@cemm.at) , Enrico Girardi (EGirardi@cemm.oeaw.ac.at)

Review Timeline:

Submission Date:	22nd Apr 20
Editorial Decision:	26th May 20
Revision Received:	16th Jun 20
Editorial Decision:	22nd Jun 20
Revision Received:	24th Jun 20
Accepted:	26th Jun 20

Editor: Maria Polychronidou

Transaction Report:

26th May 2020

Manuscript Number: MSB-20-9652, A substrate-based ontology for human Solute Carriers

Thank you again for submitting your work to Molecular Systems Biology. We have now heard back from the three referees who agreed to evaluate your study. Overall, the reviewers think that the study presents a potentially valuable tool for SLC characterization. They raise however a series of concerns, which we would ask you to address in a revision.

I think that the reviewers' recommendations are rather clear and there is therefore no need to repeat the points listed below. Please let me know in case you would like to discuss any of the issues raised. As reviewer #1 mentions, analyses in further cell lines (in addition to MCF7 cells) would more robustly support the presented conclusions and would enhance their overall significance.

REFEREE REPORTS

Reviewer #1:

In this study, the authors classify the SLCs based on a variety of features including substrate type, transport mechanism, and expression patterns in combination with machine learning. This study represents a creative and promising way to characterize this highly understudied gene family, and provides a unique approach to deorphanize many SLC members. I have several comments that can improve the manuscript.

- 1) The description of the machine learning algorithm is very thin. More information about the method performance and evaluation is needed. For example, how was it trained? How were the parameters optimized? Did they use other algorithms?
- 2) They analyzed expression data from the MCF7 cell line after nutrient deprivation. While this data is very interesting, it would be useful to look at other cell lines as well. This will help eliminate artifacts, provide another application of the approach, and may reveal new biology.
- 3) How was the transport mechanism (e.g. symporter, antiporter) annotated? Some transporters can have more than one mechanism, for example.

4) Previous attempts have been made to look at some versions of these features in SLCs and they should be mentioned (e.g., Schlessinger et al, Protein Sci. 2010). Recent study by the Giacomini lab has also deorphanized an SLC22 transporter based on various orthogonal approaches (Yee et al, PLOS Genetics 2019). A deeper discussion on this topic should be added.

Reviewer #2:

Solute carriers (SLCs) are a large family of integral membrane proteins that facilitate the transport of a variety of substrates. Although SLCs are of high clinical importance, we lack a thorough understanding of their substrates and metabolic function. In this manuscript Meixner et. al. describe a manual annotation of the Solute-Carrier protein family where they have performed a literature based manually curated analysis in which they describe the known substrates, transporter mechanism and subcellular locations of known SLCs. They further demonstrate the applicability of this dataset for a more specific term enrichment analysis of transcriptional data obtained from a study of MCF7 breast cancer cells under single amino acid starvation conditions. By implicating a machine learning model the authors demonstrate a prediction of substrates for the 124 orphan SLCs. The authors have established an online interactive version of this manually curated annotation for the scientific community. This work is of value to the scientific community in facilitating a high-confidence organized resource for SLC substrate ontology that will enable to further perform substrate-transporter based inter-relationship studies, for example between different cell and tissue types. The significance of aberrant functioning of SLCs in diseases further signifies the importance of such a resource not only to systems biology community but also for a wider audience in drug development. The authors have pioneered previously significant contributions in the field of SLC biology and this work. Before I can recommend this manuscript to be accepted for publication in Molecular Systems Biology the authors should address the following comments and suggestions:

In principle the manuscript is structured and communicated clearly. Nevertheless, I would suggest to reorganize Figure 1 to improve the flow and following through the text body.

Figure 3A, 3B, EV1 contain red boxes in the figure but the figure legend nor the text body indicates what the highlighted red boxes exactly represent.

In Figure 3A/EV2 there is a group of transporters for which the predicted normalized class probability is relatively low. Can the authors further elaborate on this in the results and/or discussion section? Is there any common characteristic for this group of SLCs in this observation?

SLCs are known targets for several drugs. Would it be possible to predict additional specificities for these drugs based on the substrate prediction performed for orphan SLCs? It would be very interesting to see how applying this substrate ontology data would work in predicting additional SLCs that could potentially use drugs as transport substrates - especially in the orphan SLC family.

Reviewer #3:

Solute carriers (SLCs) are a superfamily of transporters that, relative to other superfamilies, are poorly characterized. This dearth of characterization spans all biochemical and functional aspects,

however, lack of consistent and broad substrate annotation is particularly prominent. In this manuscript, the authors describe a manual-curated ontology of human SLCs based on their known substrates and primary literature. The authors subsequently demonstrated the utility of this ontology by applying it to characterize transcriptomics data and also to train a machine learning model to make SLC substrate predictions for orphan SLCs. Considering the massive deficiency SLC substrate characterization, the work herein represents a valuable tool for various aspects of SLC research. Thus, this manuscript is certainly qualified to be published in *Molecular System Biology* and I believe will be of great interest to the readership and more broadly to the community. Further, the authors have already made this ontology public and available to users, delivering a valuable resource to the community. However, I do have some thoughts as well as some more minor comments that I believe the authors should address prior to publication.

1. I'm not sure whether the hierarchical term directly taken from ChEBI will faithfully group SLC substrates (and therefore SLCs) based on their structure and physiochemical differences, which seems to be consistent main goal of the authors of connecting substrate/function to other parameters (e.g. sequence, structure, etc). For example, in Figure 2E, 'Glycine' and 'L-serine' belong to the same ChEBI term 'Serine family aa', which are all amino acids biosynthesized from 3-phosphoglycerate. However, 'L-serine' is more chemically related to 'L-threonine' than 'glycine'. In another example, isoleucine (a neutral hydrophobic amino acid) is grouped with asparagine (neutral polar), aspartic acid (negatively charged) and lysine (positively charged), etc, since they are all derived from aspartate, however, chemically are very different. It seems subcategory terms that more genuinely reflect structural/chemical features of substrates could be more accurate for orphan SLC substrate predication.

2. I'm a bit confused, in the results section there is a sentence "We found that for 126 SLCs no substrates were experimentally confirmed to be transported in human cells..." Based off of the methods section, it is unclear as to whether the manual curation by the authors restricted to human SLCs, or also included non human SLCs? Further, the authors specifically mention "human cells", however, their methods indicate that substrates are "defined as molecules showed to be transported by the SLC in a transport assay using reconstituted protein or gene overexpression experiments." So, it does not seem restricted to data "confirmed to be transported in human cells." The authors should be more specific here in the results section and the methods section about specific species search and whether the search was restricted to cells or not.

Minor:

1. In the abstract, the sentence below makes it sound like ~30% of SLCs lack substrates, which they obviously do not, but rather lack "known" substrates...

"The annotation of substrates, transport mechanism, coupled ions and subcellular localization for 446 human SLCs confirmed that ~30% of these were still functionally orphan and lacked substrates."

2. In Figure 1C, the two substrate classes 'ion' and 'metal' seem to overlap, it might be necessary to explain the exact meaning and inclusion criteria for 'ion' and metal in figure 1 legend. In Figure 1D, there is category named 'symporter; antiporter' and 'uniporter; antiporter', do they mean SLCs that manifested two kinds of mechanisms for different substrates?

3. How do the authors annotate SLC localization (Fig1B)? Their parameters for localization are not defined.

A substrate-based ontology for human Solute Carriers (MSB-20-9652)

Meixner et al

Point-by-point response to the reviewers' comments

Reviewer #1:

In this study, the authors classify the SLCs based on a variety of features including substrate type, transport mechanism, and expression patterns in combination with machine learning. This study represents a creative and promising way to characterize this highly understudied gene family, and provides a unique approach to deorphanize many SLC members. I have several comments that can improve the manuscript.

1) The description of the machine learning algorithm is very thin. More information about the method performance and evaluation is needed. For example, how was it trained? How were the parameters optimized? Did they use other algorithms?

To clarify the training procedure, we added the following sentence to the Methods section:

"In total, 18 substrate terms were selected from the SLC substrate ontology which are generic enough to have a sufficient number of annotated SLCs for training as well as specific enough to allow hypothesis generation by subsequent prediction on orphan SLCs. For each substrate term, an independent, binary classifier was trained on the set of 304 SLCs with known substrates and sequence features, using those SLCs with a known substrate matching to the specific ontology term as a positive training set and the remaining SLCs with a known substrate as a negative training set." We now also list the range used to optimize the parameters used in our method.

We decided on Random Forest as machine learning algorithm as it is rather insensitive to feature selection and provides convenient out-of-bag estimates. Also, it has been successfully used for protein sequence classification before (Bausch-Fluck et al., 2018). We did not do a comparison of machine learning algorithms as part of this study.

2) They analyzed expression data from the MCF7 cell line after nutrient deprivation. While this data is very interesting, it would be useful to look at other cell lines as well. This will help eliminate artifacts, provide another application of the approach, and may reveal new biology.

We thank the reviewer for its suggestion. We now included a new transcriptomics dataset obtained in our laboratory by removing specific amino acids from the media of HEK293T cells (Fig3A, Fig EV2, EV3). Analysis of this dataset revealed a broad similarity between the two cell lines, confirming the robustness of our analysis. We modified the Methods and the main text (page 5, section "Application of the substrate ontology for term enrichment analysis in transcriptomic data")

3) How was the transport mechanism (e.g. symporter, antiporter) annotated? Some transporters can have more than one mechanism, for example.

Transport mechanism was annotated accordingly to the corresponding primary literature. Some transporters can indeed have multiple mechanisms and this is reflected by the presence of multiple

mechanism annotations for a subset of SLCs, as shown in figure 1, which includes several members of the SLC1 and SLC38 families.

4) Previous attempts have been made to look at some versions of these features in SLCs and they should be mentioned (e.g., Schlessinger et al, Protein Sci. 2010). Recent study by the Giacomini lab has also deorphanized an SLC22 transporter based on various orthogonal approaches (Yee et al, PLOS Genetics 2019). A deeper discussion on this topic should be added.

We thank the reviewer for pointing this out. We now included the mentioned references and the importance of orthogonal approaches in the Discussion section (page 7).

Reviewer #2:

Solute carriers (SLCs) are a large family of integral membrane proteins that facilitate the transport of a variety of substrates. Although SLCs are of high clinical importance, we lack a thorough understanding of their substrates and metabolic function. In this manuscript Meixner et. al. describe a manual annotation of the Solute-Carrier protein family where they have performed a literature based manually curated analysis in which they describe the known substrates, transporter mechanism and subcellular locations of known SLCs. They further demonstrate the applicability of this dataset for a more specific term enrichment analysis of transcriptional data obtained from a study of MCF7 breast cancer cells under single amino acid starvation conditions. By implicating a machine learning model the authors demonstrate a prediction of substrates for the 124 orphan SLCs. The authors have established an online interactive version of this manually curated annotation for the scientific community. This work is of value to the scientific community in facilitating a high-confidence organized resource for SLC substrate ontology that will enable to further perform substrate-transporter based inter-relationship studies, for example between different cell and tissue types. The significance of aberrant functioning of SLCs in diseases further signifies the importance of such a resource not only to systems biology community but also for a wider audience in drug development. The authors have pioneered previously significant contributions in the field of SLC biology and this work. Before I can recommend this manuscript to be accepted for publication in Molecular Systems Biology the authors should address the following comments and suggestions:

In principle the manuscript is structured and communicated clearly. Nevertheless, I would suggest to reorganize Figure 1 to improve the flow and following through the text body.

We thank reviewer for pointing out the issue. We rearranged panels in Figure 1 to make it corresponds to the flow of the text.

Figure 3A, 3B, EV1 contain red boxes in the figure but the figure legend nor the text body indicates what the highlighted red boxes exactly represent.

We thank the reviewer for pointing this out. The red boxes represent significant enrichments. We updated the figure legend accordingly.

In Figure 3A/EV2 there is a group of transporters for which the predicted normalized class probability is relatively low. Can the authors further elaborate on this in the results and/or discussion section? Is there any common characteristic for this group of SLCs in this observation?

We thank the reviewer for this comment. Indeed, there appears to be a block of 19 orphan SLCs with overall rather low probabilities for any of the classifiers. We could not find any common characteristic within those SLCs. The explanation is, that we do not aim for an exhaustive (complete) classification here, as we only have sufficient training data for a small number of substrate terms. The following paragraph has been added to the Methods section to emphasize this point:

“In total, 18 substrate terms were selected from the SLC substrate ontology which are generic enough to have a sufficient number of annotated SLCs for training as well as specific enough to allow hypothesis generation by subsequent prediction on orphan SLCs.”

SLCs are known targets for several drugs. Would it be possible to predict additional specificities for these drugs based on the substrate prediction performed for orphan SLCs? It would be very interesting to see how applying this substrate ontology data would work in predicting additional SLCs that could potentially use drugs as transport substrates - especially in the orphan SLC family.

Our group has previously extensively explored the associations between SLCs and drugs using both *in silico* (César-Razquin et al, *Frontiers in Pharmacology*, 2018) and experimental approaches (Girardi et al, *Nature Chemical Biology*, 2020) but the number of drugs targeting SLCs is still relatively low (César-Razquin et al, *Cell*, 2015), precluding their use in training a Random Forest classifier. More throughout analyses, beyond the scope of the current manuscript, will be likely required to address this point. However, we predict that already in a few years the fast-growing number of SLC-targeting compounds will enable a successful training of the model.

Reviewer #3:

Solute carriers (SLCs) are a superfamily of transporters that, relative to other superfamilies, are poorly characterized. This dearth of characterization spans all biochemical and functional aspects, however, lack of consistent and broad substrate annotation is particularly prominent. In this manuscript, the authors describe a manual-curated ontology of human SLCs based on their known substrates and primary literature. The authors subsequently demonstrated the utility of this ontology by applying it to characterize transcriptomics data and also to train a machine learning model to make SLC substrate predictions for orphan SLCs. Considering the massive deficiency SLC substrate characterization, the work herein represents a valuable tool for various aspects of SLC research. Thus, this manuscript is certainly qualified to be published in *Molecular System Biology* and I believe will be of great interest to the readership and more broadly to the community. Further, the authors have already made this ontology public and available to users, delivering a valuable resource to the community. However, I do have some thoughts as well as some more minor comments that I believe the authors should address prior to publication.

1. I'm not sure whether the hierarchical term directly taken from ChEBI will faithfully group SLC substrates (and therefore SLCs) based on their structure and physiochemical differences, which seems to be consistent main goal of the authors of connecting substrate/function to other parameters (e.g. sequence, structure, etc). For example, in Figure 2E, 'Glycine' and 'L-serine' belong to the same ChEBI term 'Serine family aa', which are all amino acids biosynthesized from 3-phosphoglycerate. However, 'L-serine' is more chemically related to 'L-threonine' than 'glycine'. In another example, isoleucine (a neutral hydrophobic amino acid) is grouped with asparagine (neutral polar), aspartic acid (negatively charged) and lysine (positively charged), etc, since they are all derived from aspartate, however, chemically are very different. It seems subcategory terms that more genuinely reflect structural/chemical features of substrates could be more accurate for orphan

SLC substrate predication.

In ChEBI, and therefore also in the derived SLC substrate ontology, chemical compounds are grouped by a number of different aspects, including structural aspects, physicochemical aspects as well as metabolic aspects. The example of an ontology sub-graph we show in Figure 2D is based on the term “L- α amino acid”, where connected sub-terms are metabolic related. Regarding the examples brought up by the reviewer, L-serine and L-threonine are also child terms to the substrate term “polar amino acids” and L-isoleucine is also child term to the substrate term “branched-chain amino acid”. All of these terms are connected to the term “amino acid” at some point higher up in the ontology, but for readability, in Figure 2D we focused on the rather small sub-graph of “L- α amino acid”.

We added a new supplemental figure EV1 which shows the much bigger sub-graph for the general “amino acid” term. Here the connections described above become visible.

In addition, we added the following sentence to the figure legend of Fig 2D:

“This is a sub-graph and SLC substrates (grey) are connected to more terms in the full ontology. Please refer to Fig EV1 for an extended example of the term amino acid and its sub-terms.”

2. I'm a bit confused, in the results section there is a sentence "We found that for 126 SLCs no substrates were experimentally confirmed to be transported in human cells..." Based off of the methods section, it is unclear as to whether the manual curation by the authors restricted to human SLCs, or also included non human SLCs? Further, the authors specifically mention "human cells", however, there methods indicate that substrates are "defined as molecules showed to be transported by the SLC in a transport assay using reconstituted protein or gene overexpression experiments." So, it does not seem restricted to data "confirmed to be transported in human cells." The authors should be more specific here in the results section and the methods section about specific species search and whether the search was restricted to cells or not.

The manual curation was restricted to human SLCs and transport assays or experiments performed in human cells. We modified the Methods section to clarify this point.

Minor:

1. In the abstract, the sentence below makes it sound like ~30% of SLCs lack substrates, which they obviously do not, but rather lack "known" substrates...

"The annotation of substrates, transport mechanism, coupled ions and subcellular localization for 446 human SLCs confirmed that ~30% of these were still functionally orphan and lacked substrates."

Many thanks for pointing this out. We modified the abstract text accordingly.

2. In Figure 1C, the two substrate classes 'ion' and 'metal' seem to overlap, it might be necessary to explain the exact meaning and inclusion criteria for 'ion' and metal in figure 1 legend. In Figure 1D, there is category named 'symporter; antiporter' and 'uniporter; antiporter', do they mean SLCs that manifested two kinds of mechanisms for different substrates?

In the SLC annotation, we now renamed the 'metal' class to 'heavy metals' (defined as one of Cd²⁺, Co²⁺, Cu²⁺, Fe²⁺, Mn²⁺, Ni²⁺, Pb²⁺, V³⁺, VO²⁺, Se²⁺, Zn²⁺) and updated figures and text accordingly. The 'ion' class refers to small charged molecules and atoms, excluding heavy metals. We now expanded the corresponding Methods section and we would refer to the full annotation table for the exact allocation of each molecular species to one of the two substrate classes.

As mentioned in our response to reviewer #1, several SLCs indeed exhibit more than one transport mechanism, depending on the substrate transporters and this is reflected in the dual annotation presented in Fig1 and in the annotation table.

3. How do the authors annotate SLC localization (Fig1B)? Their parameters for localization are not defined.

SLC localization was annotated based on available immunofluorescence data obtained in human cells. Whenever conflicting information was available, precedence was given to localization data obtained by co-staining with known organelle markers. We modified the relevant Methods section to clarify this point.

22nd Jun 2020

Manuscript Number: MSB-20-9652R

Title: A substrate-based ontology for human Solute Carriers

Thank you for sending us your revised manuscript. We think that the performed revisions satisfactorily address the issues raised by the reviewers. As such, I am glad to inform you that your manuscript is now suitable for publication, pending some minor editorial issues listed below.

Corresponding Author Name: Giulio Superti-Furga

Manuscript Number: MSB-20-9652